# Notch Signaling Pathway in Pancreatobiliary Tumors

**DOI:** 10.3390/medicina57020105

**Published:** 2021-01-24

**Authors:** Francesca Borlak, Anja Reutzel-Selke, Anja Schirmeier, Julia Gogolok, Ellen von Hoerschelmann, Igor M. Sauer, Johann Pratschke, Marcus Bahra, Rosa B. Schmuck

**Affiliations:** Department of Surgery, Experimental Surgery, Campus Charité Mitte, Campus Virchow Klinikum, Charité–Universitätsmedizin Berlin, Corporate Member of Freie Universität Berlin, Humboldt-Universität zu Berlin, and Berlin Institute of Health, 10117 Berlin, Germany; francesca.borlak@charite.de (F.B.); anja.reutzel-selke@charite.de (A.R.-S.); anja.schirmeier@charite.de (A.S.); julia.gogolok@charite.de (J.G.); Ellen.Hoerschelmann@charite.de (E.v.H.); igor.sauer@charite.de (I.M.S.); johann.pratschke@charite.de (J.P.); marcus.bahra@charite.de (M.B.)

**Keywords:** pancreatic cancer, cholangiocarcinoma, cancer stem cells, notch pathway

## Abstract

*Background and Objectives:* The Notch signaling pathway plays an important role both in the development of the ductal systems of the pancreas and the bile ducts as well as in cancer development and progression. The aim of this study was to examine the expression of central proteins of the Notch signaling pathway in pancreatobiliary tumors and its influence on patient survival. *Materials and Methods:* We compared the receptors (Notch1, Notch4), activating splicing factors (ADAM17), and target genes (HES1) of the Notch pathway and progenitor cell markers with relevance for the Notch signaling pathway (CD44, MSI1) between pancreatic adenocarcinomas (PDAC, *n* = 14), intrahepatic cholangiocarcinoma (iCC, *n* = 24), and extrahepatic cholangiocarcinoma (eCC, *n* = 22) cholangiocarcinomas via immunohistochemistry and ImageJ software-assisted analysis. An Immunohistochemistry (IHC)-score was determined by the percentage and intensity of stained (positive) cells (scale 0–7) and normal and malignant tissue was compared. In the IHC results, patients’ (gender, age) and tumor (TNM Classification of Malignant Tumors, Union Internationale contre le Cancer (UICC) stages, grading, and lymphangitic carcinomatosa) characteristics were correlated to patient survival. *Results:* For eCC, the expression of CD44 (*p* = 0.043, IHC-score 3.94 vs. 3.54) and for iCC, the expression of CD44 (*p* = 0.026, IHC-score 4.04 vs. 3.48) and Notch1 (*p* < 0.001, IHC-score 2.87 vs. 1.78) was significantly higher in the tumor compared to non-malignant tissue. For PDAC, the expression of ADAM17 (*p* = 0.008, IHC-score 3.43 vs. 1.73), CD44 (*p* = 0.012, IHC-score 3.64 vs. 2.27), Notch1 (*p* = 0.012, IHC-score 2.21 vs. 0.64), and Notch4 (*p* = 0.008, IHC-score 2.86 vs. 0.91) was significantly higher in the tumor tissue. However, none of the analyzed Notch-signaling related components showed an association to patient survival. *Conclusion:* A significant overexpression of almost all studied components of the Notch signaling pathway can be found in the tumor tissue, however, without a significant influence on patient survival. Therefore, further studies are warranted to draw conclusions on Notch pathway’s relevance for patient survival.

## 1. Introduction

Pancreatic ductal adenocarcinoma (PDAC) and cholangiocellular carcinoma (CC) are both characterized by a high mortality rate and poor prognosis [1]. CCs are sub-classified into intrahepatic cholangiocarcinoma (iCC) and extrahepatic cholangiocarcinoma (eCC), the latter comprising hilar (hCC) and distal (dCC) cholangiocarcinomas [2]. PDAC and CC both share clinical and molecular features, especially tumors of the pancreatic head and dCCs are often difficult to distinguish [3,4]. Both tumor entities reside in the ductal system of the pancreas or bile ducts respectively. Those ductal structures arise from the ventral foregut over the course of embryological development [5]. During this development, as well as in adult tissue maintenance, the Notch signaling pathway plays a pivotal role [6]. When a Notch ligand such as Jagged binds to the Notch receptor of the target cell, ADAM metalloproteases cleave the ligand-binding part of the Notch receptor, while the intracellular domain of the Notch receptor (NICD) is released. NICD reaches the nucleus of the target cell via nuclear pores and binds to the transcription factor CSL. The new complex of NID and CSL now has an activating effect on the transcription level and leads to an increased expression of important target genes [7]. Thereby the Notch signaling pathway leads to an inhibition of differentiation processes such as vascular and neuronal structures and preserves the progenitor state of cells. It is therefore not surprising that its important role in cancer stem cells (CSC) is well established [8]. Dysregulations of the Notch pathway can be found in various tumor entities such as ovary, prostate, bladder, and colon cancer [9]. The role of the Notch signaling pathway in pancreatic ductal adenocarcinoma [10] and cholangiocarcinoma [11] is of particular interest. For instance, studies show that Notch receptors such as Notch1 are overexpressed in PDAC [12], with in vitro studies pointing at a reduction of tumor-promoting factors by e.g., Notch4 inhibition [13]. An overexpression of Notch1 and Notch4 can also be found in iCC and eCC, correlating with tumor aggressiveness [14,15,16]. The activation of this evolutionarily highly conserved pathway enhances the malignant features of the tumor in various ways. Firstly, epithelial to mesenchymal transition (EMT) is promoted by Notch activation in a contact dependent matter. EMT is a fundamental mechanism in invasion and metastasis as it allows the dissemination of cells from the primary tumor and therefore triggers local and systemic spread [17,18]. It has also been shown to induce resistance to conventional therapeutics with high clinical relevance in PDAC and CC such as paclitaxel and oxaliplatin [19]. Moreover, other mechanisms of chemoresistance have been described to be induced via the Notch pathway such as autophagy and down-regulation of E-cadherin to name two examples [20,21]. These findings imply the potential of Notch inhibition as a targeted approach in cancer therapy.

The aim of this study was to investigate the expression of compounds of the Notch signaling pathway in both PDAC and CC and evaluate their effect on patient survival.

## 2. Materials and Methods

Patients included in this study underwent resection of a tumor of the pancreas or bile ducts respectively. Surgical techniques included partial or complete resection of the pancreas, extrahepatic resection of the bile duct, and partial liver resection. Samples of tumor tissue and non-malignant tissue were obtained right after resection by an experienced pathologist to avoid impairment of clinical diagnostics. The final diagnosis was verified by histological assessment and additional immunohistochemistry, if needed, by a senior pathologist specialized in pancreatobiliary malignancies. All patients gave written permission and the study was approved by the local ethics committee (EA1/292, EA2/035). All analyses were conducted according to the Helsinki declaration of 1975, as revised in 1983. Patients younger than 18 years were excluded from this study. Patient data, clinical follow up, and pathological evaluation including TNM Classifications of Malignant Tumors and Union Internationale contre le Cancer (UICC) stages were collected for further analysis.

Samples were cryoconservated in liquid nitrogen and stored at −80 °C until analysis. A total of 30 µm of sections were generated with a cryostat at −20 °C and samples were fixed. Immunohistochemical staining was conducted using the following monoclonal mouse (CD44, eBioscience#BMS150) and polyclonal rabbit primary antibodies: MSI (Antibodies #ABIN953550), Notch1 (Cell Signaling #D6F11), Notch4 (Avivasysbio #ARP32052-P050), ADAM17 (Abgent #AP1492a), and HES1 (St. John’s #STJ23938). The biotin-horseradish peroxidase linked LSAB system (Dako #K0-690) was used for visualization. Five visual fields were captured with a Keyence microscope (Osaka, Japan). All samples were stained as a set with one antibody. A semi-quantitative score (IHC score) was applied for interpretation including both intensity of the staining (0 to 4) and percentage of positive cells (0 to 3) resulting in a count from 0 to 7.

For statistical analysis, SPSS (Superior Performing Software System) was used (version 16 and 22, IBM, Germany). Continuous data were expressed as median with range and categorical data as frequency and percentage in brackets. Comparisons between groups were analyzed with the Kruskal–Wallis test. Means of the immunohistochemical scores were compared with the Wilcoxon test. To compare the effect of antigen expression, the difference between non-malignant tissue and tumor tissue was analyzed and displayed in three groups: More, equal, or less antigen expression in the tumor compared to non-malignant tissue. The Log Rank (Mantel–Cox) method was used to assess cumulative survival that was plotted as Kaplan–Meier curves. A *p*-value equal or less than 0.05 was considered to be statistically significant.

## 3. Results

### 3.1. Patient and Tumor Characteristics

A total of 60 patients were included in the study: 14 suffering from PDAC, 22 from eCC, and 24 from iCC. None of the patients received neoadjuvant therapy before surgery. Patients were compared in terms of gender, age, survival, and UICC stage. More female than male patients were included in the study. The median age was between 60.5 and 65 years across the three tumor entities. The difference regarding gender and age was not significant. Regarding tumor characteristics we found significant differences in the size of the tumor (T stage) and lymphangitic carcinomatosa (LC) between the tumor entities with significantly bigger tumors and more LC in PDAC (Table 1).

### 3.2. Comparison between Tumor and Non-Malignant Tissue

A higher expression of signal proteins of the Notch pathway was found in malignant tissue (T) compared to normal tissue (N). Yet this proofed not always statistically significant. For eCC, the expression of CD44 (*p* = 0.043), for iCC the expression of CD44 (*p* = 0.026), and Notch1 (*p* < 0.001), and for PDAC the expression of ADAM17 (*p* = 0.008), CD44 (*p* = 0.012), Notch1 (*p* = 0.012), and Notch4 (*p* = 0.008) was significantly higher in malignant tissue. PDAC showed the highest rate of Notch components overexpressed in the tumor tissue. Average IHC scores of normal and malignant tissue and respective *p*-values are displayed in Table 2, representative staining in Figure 1. Regarding the location of the staining, ADAM17 and CD44 could be detected in cytoplasm und cell membranes, HES1 and MSI in the cell nuclei and cytoplasm and Notch1 and Notch4 in the cytoplasm.

### 3.3. Effect of Expression of Notch Pathway on Patient Survival

None of the analyzed Notch-signaling related components showed an association to patient survival (Table 3). Representative Kaplan–Meier curves are displayed in Figure 2. However, specific patient and tumor characteristics showed an association to patient survival: In iCC and PDAC, UICC stage had an effect on patient survival. In iCC, a larger tumor size showed a negative effect on survival, and in PDAC this was true for the occurrence of metastasis. Furthermore, the grading of the tumor had an effect on survival if all tumor entities were cumulatively analyzed.

## 4. Discussion

In this study we analyzed the antigen expression of components of the Notch signaling pathway in pancreatobiliary tumors. We could detect a higher expression in tumor tissue compared to non-malignant tissue for the majority of the analyzed components. We investigated both receptors (Notch1 and Notch4), activating proteases (ADAM17), target genes (HES1), and progenitor cell markers with relevance for the Notch signaling pathway (CD44 and MSI1).

In mammals, four Notch receptors have been described (Notch1–4) of which Notch1 and Notch4 are of particular interest in solid tumors. All Notch receptors are highly expressed during the development of the pancreas and bile ducts and very seldom detectable in adult tissue Notch [22]. However, both Notch1 and Notch4 have also been described to be highly re-expressed in pancreatic cancer and cholangiocarcinomas [12,13,14,15,16]. This can be seen as a sign of reactivated embryologic pathways in the respective cancers. This activation can fittingly also be found in cancer stem cells [23]. In our cohort, we could prove a higher expression of Notch receptors in iCC (Notch1) and PDAC (Notch1 and Notch4), however, without showing an influence on survival. A comparable study by Song et al. showed an overexpression of Notch1 in PDAC without a significant effect on patients survival [12]. Furthermore, regarding iCC and eCC, studies could not detect a significant overexpression of Notch4 compared to normal tissue which matches our results [24,25]. In an attempt at focused Notch4 receptor inhibition, Qian et al. could detect an inhibition of migration and invasion of PDAC cells and induced chemoresistance in vitro.

The activation of the Notch pathway is induced by disconnecting the intra- and extracellular fraction of the receptor. This process is carried out by the metalloprotease ADAM17. It has already been shown to be overexpressed in eCCs with a negative effect on survival in a study by Jiao et al. [26]. The fact that it is already overexpressed in precancerous lesions in the pancreas points to its role in tumor initiation [27]. In contrast to the latter study, we also found a weak expression of ADAM17 in non-malignant pancreatic tissue, however, significantly less expressed than in the studied tumor tissue. In vivo, the progression of pre-invasive pancreatic lesions to advanced PDAC could be blocked by an ADAM17 antibody in mice [28]. ADAM17 inhibitors (e.g., Aderbasib/INCB7839) have also already been under clinical investigation for various cancer types including lung and ovarian cancer with promising results [29,30].

The target gene of the Notch pathway HES1 is also commonly overexpressed in premalignant lesions as well as tumors of the pancreas and bile ducts [31,32] and is furthermore associated with poor survival. Interestingly, an increased expression of HES1 triggered by pancreatic stellate cells leads to a resistance to gemcitabine, a drug with high relevance in the therapy of PDAC and bile duct cancer [33]. We could only detect a trend regarding overexpression in eCC (*p* = 0.062). However, an association to patient survival could not be shown. Aoki et al. could show the same effect in their study in eCC.

Pancreatic stellate cells (also referred to as cancer associated fibroblasts/ CAFs in tumors) form a large proportion of the tumor microenvironment. The bidirectional interaction between tumor cells and tumor microenvironment plays an important role in cancer progression. The Notch pathway regulates this interaction and therefore shapes the tumor microenvironment. For example, CAFs induce Notch activation and conversely, induction of Notch3 by CAFs is associated with an increase in cancer stem cells in hepatocellular carcinoma [34,35]. Furthermore, Notch leads to an activation of CAFs leading to an inflammatory phenotype [36].

Besides analyzing the described Notch signaling pathway components, we investigated the two cancer stem cell markers CD44 and MSI. Both have been described to play a role in the activation of the Notch signaling pathway [37,38]. Cancer stem cells themselves have been established to play a pivotal role in chemoresistance and have a high tumor-initiating potential due to their potential of sustained self-renewal, proliferation, and differentiation. We could not detect an overexpression of MSI in the analyzed tumors, but we did show a significant overexpression of CD44 in all three tumor entities. This overexpression has also been described previously in tumors of the pancreas and bile ducts [39,40]. CD44 is a transmembrane glycoprotein that is a downstream target of the Wnt/B-catenin pathway and serves as a receptor for hyaluronic acid. It plays important roles in cell migration, differentiation, and survival signaling in both PDAC and CC [41,42].

When comparing expression patterns of the three tumor entities, PDAC showed the highest rate of Notch components being overexpressed in the tumor tissue (4 out of 6), iCC and eCC showed an overexpression of 2 or 1 components respectively. As to our knowledge, in the existing literature there is no such comparison between these three tumor entities in regard to signal proteins of the Notch signaling pathway.

In summary, not all studied variables had a significant effect on patient outcome. UICC, T, and G had a significant effect on patient survival, however, not in all tumor entities studied. This can may be be explained by the small sample sizes in the groups (e.g., only 2 patients with UICC II stage in iCC). This is also a probable reason for the fact that there was no association between the activation of Notch signaling pathway and patient survival.

## 5. Conclusions

We could detect an overexpression of almost all studied components of the Notch signaling pathway with a predominance in PDAC. However, there was no association to patient survival. Therefore, further studies with a larger collective are warranted to also allow a profound comparison between the tumor entities.

## Figures and Tables

**Figure 1 medicina-57-00105-f001:**
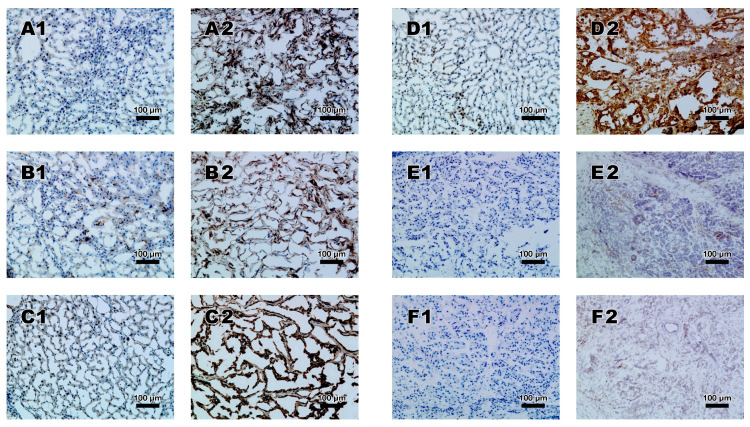
Representative antibody staining of ADAM17 (**A**), CD44 (**B**), HES1 (**C**), MSI (**D**), Notch1 (**E**), and Notch4 (**F**). Non-malignant tissue displayed on the left side (**1**), tumor tissue displayed on the right side (**2**) at a magnification of ×200.

**Figure 2 medicina-57-00105-f002:**
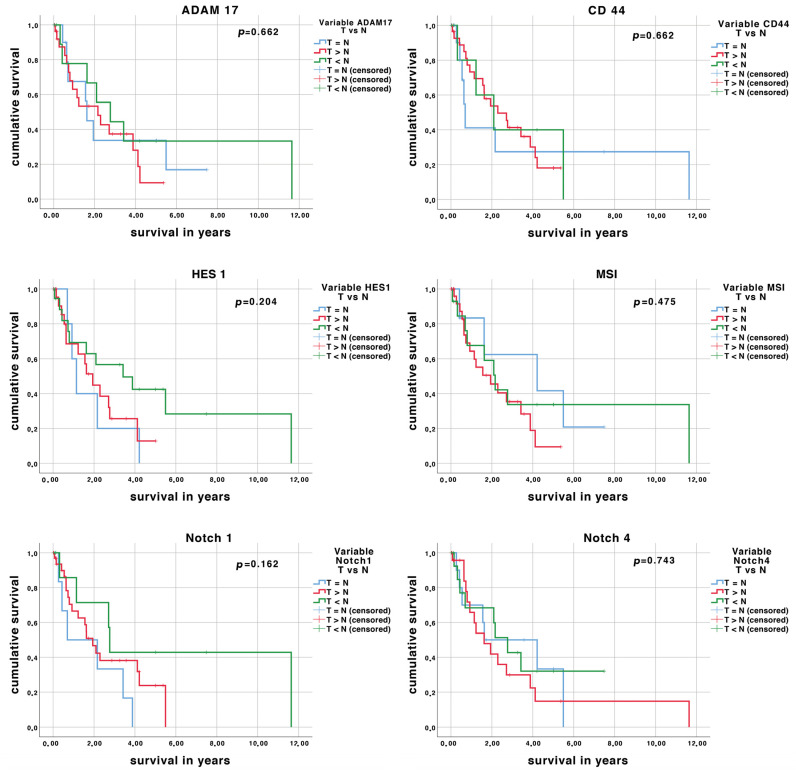
Effect of Notch components on patient survival (years) displayed as Kaplan–Meier curves. The expression of the antigens was grouped as equal (=), more (>), or less (<) antigen expression in tumor tissue (T) compared to non-malignant tissue (N). Number of patients: *n* = 60 (PDAC: 14, eCC:22, iCC:24).

**Table 1 medicina-57-00105-t001:** Comparison of patient and tumor characteristics.

Clinical Variable	iCC	eCC	PDAC	*p*-Value
	(*n* = 24)	(*n* = 22)	(*n* = 14)	
Gender				0.17
Female	19 (79)	13 (59)	7 (50)	
Male	5 (21)	9 (41)	7 (50)	
Median age (years)	65	63.5	60.5	0.524
Range (years)	39–79	41–83	43–80	
Median survival (months)	21.5	8	10.5	0.658
Range (months)	0–140	1–90	1–66	
UICC				0.638
UICC I	5 (21)	1 (5)	0 (0)	
UICC II	5 (21)	8 (36)	10 (71)	
UICC III	2 (8)	9 (41)	0 (0)	
UICC IV	12 (50)	4 (18)	4 (19)	
T				**0.011**
T1	8 (33)	2 (9)	1 (7)	
T2	7 (29)	8 (36)	0 (0)	
T3	7 (29)	10 (45)	12 (86)	
T4	1 (4)	2 (9)	0 (0)	
N				0.06
N0	14 (58)	9 (41)	3 (21)	
N1	8 (33)	11 (50)	10 (71)	
N2	0 (0)	2 (9)	0 (0)	
M				0.289
M0	19 (79)	20 (91)	9 (64)	
M1	5 (21)	2 (9)	4 (29)	
G				0.478
G0	0 (0)	2 (9)	0 (0)	
G1	1 (4)	1 (5)	0 (0)	
G2	16 (67)	13 (59)	8 (57)	
G3	5 (21)	6 (27)	5 (36)	
LC				**0.024**
yes	8 (33)	9 (41)	11 (79)	
no	16 (67)	13 (59)	3 (21)	

Note: Comparisons between groups were analyzed with the Kruskal–Wallis test. Categorical data are expressed as number (percentage) and continuous as median with range. *p* ≤ 0.05 is considered to be statistically significant. *n*: number of patients per cohort. iCC: intrahepatic cholangiocarcinoma. eCC: extrahepatic cholangiocarcinoma. PDAC: pancreatic ductal adenocarcinoma. UICC: Union internationale contre le cancer, standard for cancer staging. T: tumor size. *n*: presence of lymph node metastasis. M: presence of distant metastasis. G: Grading. LC: lymphangitic carcinomatosa. Bold numbers indicate significant results.

**Table 2 medicina-57-00105-t002:** Comparison of components the Notch signaling pathway.

Tumor Entity	N ADAM17	N CD44	N HES1	N MSI	N Notch1	N Notch4
	T ADAM17	T CD44	T HES1	T MSI	T Notch1	T Notch4
eCC	2.95	3.43	2.67	3.1	2.57	3.05
	3.5	3.94	3.44	3.72	2.83	2.78
*p*-value	0.142	**0.043**	0.062	0.1	0.472	0.629
iCC	3.17	3.48	3.09	3.3	1.78	2.96
	3.78	4.04	3.52	3.91	2.87	3.22
*p*-value	0.164	**0.026**	0.299	0.122	**0**	0.155
PDAC	1.73	2.27	4.27	2.55	0.64	0.91
	3.43	3.64	2.5	3.36	2.21	2.86
*p*-value	**0.008**	**0.012**	**0.012**	0.125	**0.012**	**0.008**

Note: Comparison was carried out with with the Wilcoxon-test. IHC scores of non-malignant (N) and tumor tissue (T) and *p*-values are given. *p* ≤ 0.05 is considered to be statistically significant. iCC: intrahepatic cholangiocarcinoma. eCC: extrahepatic cholangiocarcinoma. PDAC: pancreatic ductal adenocarcinoma. Bold numbers indicate significant results.

**Table 3 medicina-57-00105-t003:** Association of analyzed variables on patient survival.

Variables	Entities Cumulative	iCC	eCC	PDAC
Gender *p*-value	*p* = 0.431	*p* = 0.219	*p* = 0.189	*p* = 0.307
female				
median survival (months)	19.0 (1–140)	25.0 (1–140)	8.00 (1–90)	10.0 (1–46)
male				
median survival (months)	8.0 (0–66)	2.0 (0–60)	8.0 (0–50)	12.0 (1–66)
UICC *p*-value	*p* = 0.150	***p* = 0.009**	*p* = 0.113	*p* = 0.051
UICC I				
median survival (months)	8.00 (0–43)	2.0 (0–43)	-	
UICC II				
median survival (months)	12.0 (1–140)	60.0 (36–140)	2.5 (1–50)	11.5 (1–66)
UICC III				
median survival (months)	19.0 (2–90)	13.0 (7–19)	21.0 (2–90)	
UICC IV				
median survival (months)	8.5 (1–51)	21.5 (2–51)	7.5 (4–26)	7.5 (1–27)
T *p*-value	*p* = 0.249	***p* = 0.023**	*p* = 0.573	*p* = 0.700
T1				
median survival (months)	11.0 (0–43)	5.0 (0–43)	34.5 (26–43)	-
T2				
median survival (months)	33.0 (1–140)	51.0 (5–140)	7.5 (1–50)	
T3				
median survival (months)	12.0 (1–90)	20.0 (2–41)	11.0 (1–90)	10.0 (1–66)
T4				
median survival (months)	7.0 (2–25)	-	4.5 (2–7)	
N *p*-value	*p* = 0.319	*p* = 0.066	*p* = 0.598	*p* = 0.355
N0				
median survival (months)	27.5 (0–140)	30.5 (0–140)	21.0 (1–50)	46.0 (9–66)
N1				
median survival (months)	8.0 (1–90)	13.5 (2–41)	7.0 (1–90)	10.0 (1–65)
N2				
median survival (months)	20.0 (14–26)		20.0 (14–26)	
M *p*-value	*p* = 0.134	*p* = 0.785	*p* = 0.275	*p* = 0.054
M0				
median survival (months)	14.0 (0–140)	20.0 (0–140)	8.0 (1–90)	12.0 (1–66)
M1				
median survival (months)	9.0 (1–51)	39.0 (2–51)	16.5 (7–26)	7.5 (1–27)
G *p*-value	*p* = 0.017	*p* = 0.28	*p* = 0.068	*p* = 0.979
G0				
median survival (months)	17.0 (8–26)		17.0 (8–26)	
G1				
median survival (months)	35.0 (21–49)	-	-	
G2				
median survival (months)	12.0 (0–140)	21.5 (0–140)	8.0 (1–90)	9.5 (1–65)
G3				
median survival (months)	10.5 (1–66)	14.0 (1–36)	5.5 (1–33)	11.0 (1–66)
LC	*p* = 0.461	*p* = 0.507	*p* = 0.731	*p* = 0.953
yes				
median survival (months)				
no	13.0 (1–90)	24.0 (5–60)	14.0 (1–90)	10.0 (1–66)
median survival (months)				
	11.5 (0–140)	16.5 (0–140)	7.0 (1–85)	27.0 (9–46)
ADAM17 *p*-value	*p* = 0.662	*p* = 0.562	*p* = 0.977	*p* = 0.724
equal				
median survival (months)	19.0 (0–90)	21.0 (0–60)	8.0 (7–90)	19.0 (6–66)
more				
median survival (months)	10.5 (1–65)	8.0 (2–51)	14.0 (1–43)	10.5 (1–65)
less				
median survival (months)	25.0 (1–140)	25.0 (1–140)	18.5 (4–50)	
CD44 *p*-value	*p* = 0.969	*p* = 0.499	*p* = 0.766	*p* = 0.079
equal				
median survival (months)	6.5 (0–140)	3.5 (0–140)	7.0 (2–90)	-
more				
median survival (months)	21.0 (1–65)	36.5 (2–60)	17.5 (1–43)	11.0 (1–65)
less				
median survival (months)	14.0 (1–66)	8.0 (1–25)	27.0 (4–50)	-
HES1 *p*-value	*p* = 0.204	*p* = 0.162	*p* = 0.319	*p* = 0.887
equal				
median survival (months)	11.0 (0–51)	25.5 (0–51)	11.0 (1–26)	-
more				
median survival (months)	11.0 (1–60)	14.0 (1–60)	7.5 (2–43)	-
less				
median survival (months)	25.0 (1–140)	39.0 (2–140)	50.0 (4–90)	10.0 (1–66)
MSI *p*-value	*p* = 0.475	*p* = 0.093	*p* = 0.931	*p* = 0.089
equal				
median survival (months)	19.0 (1–90)	28.0 (5–51)	47.0 (4–90)	19.0 (1–66)
more				
median survival (months)	11.0 (0–65)	14.0 (0–49)	7.5 (1–43)	19.0 (6–65)
less				
median survival (months)	22.5 (1–140)	25.0 (2–140)	26.0 (4–50)	5.5 (1–10)
Notch1 *p*-value	*p* = 0.162	*p* = 0.161	*p* = 0.149	*p* = 0.904
equal				
median survival (months)	8.0 (2–46)	-	5.5 (2–26)	26.0 (6–46)
more				
median survival (months)	12.5 (0–66)	16.5 (0–60)	7.5 (1–50)	15.0 (1–66)
less				
median survival (months)	32.0 (1–140)	60.0 (2–140)	32.0 (4–90)	
Notch4 *p*-value	*p* = 0.743	*p* = 0.925	*p* = 0.614	*p* = 0.079
equal				
median survival (months)	20.5 (3–66)	21.0 (7–60)	12.5 (3–43)	-
more				
median survival (months)	10.0 (0–140)	8.0 (0–140)	8.0 (1–32)	11.0 (1–65)
less				
median survival (months)	25.0 (1–90)	32.0 (1–60)	15.0 (2–90)	-

Note: Survival was given as median in month with range (in brackets). Given significance according to Kaplan–Meier survival analysis. “-” = no data available due to only one patient in this specific category, blank = no patient in this specific category. *p* ≤ 0.05 is considered to be statistically significant. iCC: intrahepatic cholangiocarcinoma. eCC: extrahepatic cholangiocarcinoma. PDAC: pancreatic ductal adenocarcinoma. UICC: Union Internationale contre le Cancer, standard for cancer staging. T: tumor size. N: presence of lymph node metastasis. M: presence of distant metastasis. G: Grading. LC: lymphangitic carcinomatosa. Bold numbers indicate significant results.

## Data Availability

The data presented in this study are available on request from the corresponding author. The data are not publicly available due to sensible patient data.

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
