# Peer review of "Notch Signaling Pathway in Pancreatobiliary Tumors"

_medicina, 2021, doi:10.3390/medicina57020105_

Round 1
Reviewer 1 Report
In this study, Borlak et al. examined the Notch signaling axis in pancreatobiliary tumors such as PDAC, intra/extra-hepatic cholangiocarcinomas (CC) via IHCs of multiple markers including receptors (Notch1, Notch4), activating splicing factors (ADAM17) and target genes (HES1) of the cascade and CD44, MSI1 as progenitor cell markers. They analyzed the impact of different Notch players on patient survival of PDAC and CC. After all of the analyses, this study shows no significant correlation between patient survival and the members of Notch signaling pathway.
Firstly, as a descriptive one, this study holds an important position for basic cancer researchers focusing on PDAC&CC and Notch signaling cascade. Probably, genetic or pharmacologic alteration of these targets would create multiple differences in cancer cells but at the end what we count is the survival difference of patients. However, there are still some points to be improved in this manuscript:
Major Points:
- Introduction lacks a lot of information about Notch signaling and specific cancer types.
- At the beginning of result section, researchers should give better clarification for clinical variables. They should recheck the terminology of LC. According to the literature, most common usages are lymphangitic carcinomatosis or lymphangitis carcinomatosa, lymphangiosis carcinomatosa is more common for German references. On the other hand, table 1´s note should include meaning of p-values.
- In section 3.2, researchers should provide representative photos of IHCs for each analysis including ADAM17, CD44, HES1, MSI, Notch1, Notch4. Providing Notch1 and Notch4 is not enough for Figure 1. It is better to provide these pictures with scale bars. On the other hand, if authors can provide better images for stainings that would be great. For each marker, authors should also explain the expression profiles of specific cell types (immune cells or other stromal cells).
- In section 3.3, researchers should provide better images for Kaplan-Meier survival curves. Because they are not really nicely presented in Figure 2. The size of the curves could be increased.
- Researchers mentioned EMT and chemoresistance of cancer in the introduction. If these terms are having important base for the study, why can not we see any related data in the result section? At least to correlate, aggressiveness related markers of Notch would be presented with important EMT or MET markers. What is the chemo status of these patients? Researchers should also mention this in the material methods.
- If researchers are already aware of the main problem, which is very small sample size, why did they want to publish this study before reaching the higher numbers? If none of the Notch related members influence patient survival, and if it is due to sample size. What is the main message of this study? Researchers should modify this caveat through the manuscript.
Minor Points:
In the discussion, the expression profile of TMEN cells in these cancer types should be mentioned more often if we consider pancreatic cancer having highly desmoplastic stroma. On the other hand, line 162 pancreatic stellar cells should be changed. It is well accepted and “pancreatic stellate cells” are more common.
In the discussion, references should be explained in parallel with the findings. Especially for CD44, we all know that it is important for CSCs and cancer aggressiveness. Why is there no survival effect of CD44 expression profile? It should be mentioned in a better way by including relevant references.
Reviewer 2 Report
Borlak F et al., shows the dysregulated Notch signalling in human pancreaticobiliary cohorts and correlates the immunohistochemistry expression of several members of Notch signalling with patient survival. The manuscript is well written and the findings are easy to understand.
Minor comments:
- The abbreviations used in the manuscript need to define as it will be easy to relate to the text.
- Table 1: Please include extensions for UICC, T, N, M, and G, below the table. The above suggestions apply to other tables as well, when applicable.
- Figure 2: Increase the font size for graphical legends and titles and also include “n” number for each cohort
- Line 143/144: Please include additional references to support Notch 1 and Notch 4 dysregulation in PC and CC. The references included do not justify your claim.
- Line 145: Please correct the following sentence:
“The activation can fittingly also be found in cancer stem cells” instead of “The activation can fittingly also be found in stem cell” because of the reference you cited
- Line 148: Reference number 17 focuses on Notch 4 inhibition and not for pan-Notch receptor inhibition. Please correct the above sentence.
Round 2
Reviewer 1 Report
I would like to thank the authors for their careful consideration of my comments. They answered my questions very nicely and the manuscript is updated very well accordingly. I am happy to recommend this manuscript for publication in Medicina.This manuscript is a resubmission of an earlier submission. The following is a list of the peer review reports and author responses from that submission.